# Antecedents of Subjective Health among Korean Senior Citizens Using Archival Data

**DOI:** 10.3390/bs12090315

**Published:** 2022-08-30

**Authors:** Joonho Moon, Seoryeon Woo, Jimin Shim, Won Seok Lee

**Affiliations:** 1Department of Tourism Administration, Kangwon National University, Chuncheon 24341, Korea; 2Department of Tourism and Recreation, Kyonggi University, Seoul 03746, Korea

**Keywords:** senior citizens, subjective health, archival data, antecedents

## Abstract

This study aimed to investigate the determinants of subjective health among South Korean senior citizens. Secondary data for the year 2018 was used from the Senior Citizen Research Panel data collected by the Korea Employment Information Service. A total of 3822 valid observations were analyzed. The dependent variable was subjective health, and the independent variables were religion participation, social gathering participation, economic activity, food expenditure, leisure expenditure, travel frequency, and art watching frequency. Descriptive analysis, correlation matrix, and independent t-test were carried out for data analysis. Multiple linear regression analysis was employed using assets, age, and gender as control variables to test the research hypotheses. The results indicate that all the proposed attributes have a significant positive impact on the subjective health of Korean senior citizens, with implications for policy making.

## 1. Introduction

The average age of the population has become older in Korea. The aging of society has caused a negative effect: increasing social cost and decreasing productivity [1,2]. In Korea, 15.7% of the population is older than 65 years old, the aging has become more serious social issue due to the lowered birth rate [3]. Under this situation, it is remarkable how to promote the health condition of senior citizens because it is related to the overall welfare of society. Also, health condition is related to the population decline because a poor health condition increases the likelihood of death [4,5]. Such a condition leads this work to scrutinize more about the health condition in senior citizens. 

This study employs archival data to attain senior citizens’ information. Senior Citizen Research Panel data is the source of information. Prior studies have adopted the data sources and attained credible statistical inference [6,7]. This indicates that the data is trustworthy for statistical inference. As the main attribute of this research, subjective health is chosen as the dependent variable. Regarding the availability of the data and review of the literature, the candidates’ variables to account for subjective health are religion, social gathering, economic activity, food, leisure, travel, and art watching [8,9,10,11,12,13,14,15,16,17,18,19,20]. By comparing various elements as the explanatory variables for subjective health, this research is to identify more influential attributes. In summary, the aim of this work is to explore the determinants of subjective health for Korean senior citizens using secondary data. 

This research contributes to the literature by confirming the results of extant literature in the domain of Korean senior citizens. It might externally validate the results of prior literature. Plus, this research would be valuable for senior citizen policy making from the perspective of policy makers.

## 2. Review of Literature and Hypotheses Development

### 2.1. Subjective Health

Subjective health is an individual’s self-evaluated health condition and it considers both mental and physical health [21,22,23,24]. Scholars also assert that health condition is the basis for individual happiness because health is the starting point for everything [25,26,27,28]. Several studies have adopted subjective health as the focus of the investigation, highlighting its significance. For instance, Howell and Sweeny [29] and Ohlbrecht and Jellen [30] employed subjective health as the explained attribute. Low et al. [27] explored influential attributes of subjective health using data from 20 countries, and Sun et al. [23] researched the determinants of subjective health among Chinese participants. Poortinga et al. [24] studied the characteristics of subjective health during coronavirus disease in 2019. Moon et al. [31] also investigated the determinants of subjective health of Korean middle- and old-aged citizens. Based on the literature review, many studies have attempted to investigate subjective health. 

### 2.2. Determinants of Subjective Health 

Religion appears to be one of the factors determining subjective health. It is a sort of dependence on a powerful object, the belief which leads individuals to become more comfortable because it reduces concerns about future uncertainty, resulting in a healthier life [26,32,33,34]. Previous studies have demonstrated the effect of religion on health. Musick [8] showed a positive association between religion and subjective health among Caucasian and African American older people. Brown and Tierney [12] found a similar result among the Chinese. The second area is social gathering, which satisfies social desire through interaction with others [35,36]. Extant literature has shown that isolation causes negative effects on health conditions [37,38]. This suggests that interacting with others in social gatherings is essential to improving an individual’s health condition. Indeed, Lei et al. [10] demonstrated that social gathering enhances subjective health; while Sironi and Wolff [20] showed that isolation undermines an individual’s health condition. The third domain believed to affect subjective health is economic activity. Economic activity makes earning and life, in general, more energetic, as the daily routine encourages people to become more diligent [39,40,41,42]. Previous studies have demonstrated a link between economic activity and subjective health. Specifically, Dai et al. [15] showed that Chinese people who participated in economic activity were healthier than those who did not. Similarly, Martínez et al. [19] also revealed a positive effect of economic activity on subjective health among Columbian people. This study thus proposes the following research hypotheses:

**H1.** Religion participation positively affects subjective health.

**H2.** Social gathering participation positively affects subjective health.

**H3.** Economic activity positively affects subjective health.

The fourth factor under consideration is food expenditure. Nutritional food has a substantial role in building good health [43,44]. Food expenditure indirectly represents an individual’s food consumption quality, implying that higher food expenditure indicates healthier food consumption [45,46,47]. For instance, Noll and Weick [16] revealed that food expenditure positively affected subjective health in a study in Germany, and Kabir et al. [9] found similar results in Bangladesh. The fifth attribute is leisure expenditure. Scholars have claimed that leisure activity relieves stress and makes individuals happier as it enables them to escape from tedious daily life [48,49,50,51]. Noll and Weick [16] have also shown that subjective health in Germany is positively influenced by expenditure on leisure. Lee and Hwang [17] and Shams et al. [52] found that quality of leisure is positively associated with subjective health. Given the above, this study proposes further hypotheses as follows: 

**H4.** Food expenditure positively affects subjective health.

**H5.** Leisure expenditure positively affects subjective health.

Travel, which allows people to experience authentic things [53,54] is another domain that might influence subjective health. Prior studies have shown that travel brings about positive psychological effects and improvement in health [31,55,56,57]). McCabe et al. [56] disclosed that travel exerts a positive impact on subjective health. Yu et al. [58] and Zheng et al. [54] have reported positive associations between travel and subjective health as well. The last element considered in this research is art watching. Art watching is an appreciation of art pieces: music, picture, and performance. Art watching revitalizes emotion, inspires, and recharges mental energy [59,60,61,62]. A vast body of literature has documented the effect of art on subjective health. Västfjäll et al. [14] showed that music plays a significant role in the improvement of subjective health. Reynolds et al. [59] found that art appreciation promotes individual health status. Based on the above literature, the last two hypotheses are proposed:

**H6.** Travel positively affects subjective health.

**H7.** Art watching positively affects subjective health.

By comparing various elements as the explanatory variables for subjective health, this study sought to identify those attributes with the most influence. Thus, this study aimed to explore the determinants of subjective health of Korean senior citizens using secondary data. This study contributes to the literature by confirming the results of extant literature in the context of Korean senior citizens. It might in this manner externally validate the results of prior literature. Additionally, this study would be valuable from a policy-making perspective.

## 3. Methods

### 3.1. Data Collection and Variable Depiction

This study used secondary data, from the Senior citizen research panel data collected by the Korea Employment Information Service. As per the Service, a person above 45 years of age is considered a senior citizen. The data was collected using computer assisted personal interviewing method. Prior studies have adopted the same data source and obtained credible statistical inference [6,7]. The study period was 2018, which was the most recent period for which data was available. Initially, the number of participants was 7,490. After removing items with missing data, the number of valid observations for analysis was 3,822.

This study has one dependent variable, seven independent variables, and three control variables. All variables are measured by single item. The dependent variable is subjective health (SH), which is measured on a five-point scale (1 = very bad, 5 = very good). Religion participation (RP), social gathering participation (SP), and economic activity (EA) are measured as binary responses (0 = No, 1 = Yes). The unit of monthly food expenditure (FE) and leisure expenditure (LE) is 10,000 Korean won (KRW). The currency rate for 1 US dollar is approximately 1200 KRW in February 2022. The unit of travel frequency (TF) and art watching frequency (AF) is time. With regard to the control variables, Table 1 illustrates the measurement of assets (personal assets amount: unit ten thousand KRW), age (age of survey participants), and gender (0 = male, 1 = female). 

### 3.2. Data Analysis

This study carried out a descriptive analysis to attain mean, standard deviation, minimum, and maximum, correlation analysis to browse the overall relation between variables, and multiple linear regression analysis. Ordinary least square (OLS) regression analysis was performed to test the research hypotheses, which minimizes the sum of the square vertical distance between the regression line and observation points [63,64]. A p-value of 0.05 was selected as the cut-off value. Independent t-tests were conducted to scrutinize the difference in subjective health for the binary independent variables: religion participation (RP), social gathering participation (SP), and economic activity (EA). The following regression equation was used:*SH_i_* = *β*_0_ + *β*_1_*RP_i_* + *β*_2_*SP_i_* + *β*_3_*EA_i_* + *β*_4_*FE_i_* + *β*_5_*LE_i_* + *β*_6_*TF_i_* + *β*_7_*AF_i_* + *β*_8_*AS_i_* + *β*_9_*AG_i_* + *β*_10_*GE_i_* + *ε_i_*
where, *i* stands for the *i*th participant, *ε* is residual. 

## 4. Results

### 4.1. Descriptive Statistics and Correlation Matrix 

Table 2 shows the descriptive statistics. The mean of SH was 2.98, and its standard deviation was 0.88. Descriptive statistics also show the information of RP (Mean = 0.13, SD = 0.34), SP (Mean = 0.63, SD = 0.48), and EA (Mean = 0.37, SD = 0.48). The mean values of food expenditure (FE) and leisure expenditure (LE) were 425.9 thousand KRW and 51.6 thousand KRW, respectively. For annual travel frequency, the mean value was 1.18, and its standard deviation was 2.15. For art watching frequency, the mean value was 0.71 with 2.05 as standard deviation. Moreover, Table 2 presents the information of AS (Mean = 20425.93, SD = 32703.69), AG (Mean = 69.55, SD = 9.99), and GE (Mean = 0.42, SD = 0.49). 

Table 3 presents the correlation matrix. SH positively correlated with RP (r = 0.031), SP (r = 0.289), EA (r = 0.308), FE (r = 0.247), LE (r = 0.222), TF (r = 0.159), AF (r = 0.196), AS (r = 0.095), and GE (r = 0.091). EA negatively correlated with RP (r = -.031), whereas EA positively correlated with FE (r = 0.149), LE (r = 0.126), TF (r = 0.084), and AF (r = 0.071). AS positively correlated with RP (r = 0.010), SP (r = 0.086), EA (r = 0.050), FE (r = 0.136), LE (r = 0.272), TF (r = 0.107), and AF (r = 0.119). In contrast, AG negatively correlated with SH (r = −0.440), RP (r = −0.026), SP (r = −0.255), EA (r = −0.462), FE (r = −0.374), LE (r = −0.276), TF (r = −0.188), AF (r = −0.247), and AS (r = −0.066).

### 4.2. Results of Hypotheses Testing

The results of hypotheses testing are shown in Table 4. The model was statistically significant (F = 121.08, *p* < 0.05); RP (β = 0.076, *p* < 0.05), SP (β = 0.271, *p* < 0.05), and EA (β = 0.165, *p* < 0.05) were positively associated with SH. Also, SH was positively influenced by FE (β = 0.002, *p* < 0.05), LE (β = 0.004, *p* < 0.05), TF (β = 0.013, *p* < 0.05), and AF (β = 0.021, *p* < 0.05). AS (β = 0.001, *p* < 0.05) and GE (β = 0.105, *p* < 0.05) positively affected SH, whereas AG (β = −0.024, *p* < 0.05) negatively affected SH. Thus, all proposed hypotheses are supported. 

Table 5 presents the results of the independent t-test regarding binary independent variables. The mean value difference of RP for SH was 0.08 (t-value = 2.61, *p* < 0.05). Moreover, the mean value differences of SP and EA for SH were 0.53 (t-value = 25.18, *p* < 0.05) and 0.56 (t-value = 26.97, *p* < 0.05), respectively. The results support the hypotheses H1, H2, and H3.

## 5. Discussion

This study examined the antecedents of subjective health among Korean senior citizens. In order to inspect influential attributes, this research adopted seven attributes: religion, social gathering, economic activities, food, leisure, and art watching. The results indicated that religion participation, social gathering, and economic activity were noteworthy elements that affect the subjective health of Korean senior citizens. Moreover, religion participation appeared as the least influential attribute on subjective health among three elements: religion, social gathering, and economic activities. It demonstrated that the activities are worthy to enhance subjective health by offering a positive psychological effect. Economic activity exerted the strongest impact on subjective health compared with religion and social gatherings. It implies that economic activities are worthwhile for creating economic value as well as enhancing the health condition by satisfying the needs of belonging and esteem in life. We also found that food and leisure expenditure were significant attributes for better subjective health, with leisure expenditure having a larger impact than the former. In addition, art watching exerted a stronger influence than travel, and both were significant predictors of the subjective health condition of senior citizens. The results suggest that eating good food, traveling, and enjoying cultural activities are also very essential elements for senior citizens to become healthier. In addition, the results of independent t-tests showed the following mean values (RP: 2.97 vs. 3.05; SP: 2.64 vs. 3.17; EA: 2.77 vs. 3.33). It showed that the mean value is between two and three. Considering the characteristics of survey participants (senior citizens), they appraised their subjective health in relatively negative manners. 

## 6. Conclusions

This study theoretically contributes to the literature. It expanded the area of senior citizen research further into Korean senior citizens’ subjective health by exploring diverse determinants. Also, this study confirmed the findings from extant literature regarding the impact of various factors on subjective health. It provides external validity for prior research in terms of religion [8,12], social gathering [10,20], economic activity [15,19], food expenditure [9,16], leisure expenditure [17,18], travel [54,56], and art watching [11,14].

This study has several implications for policy making. First, policy makers could consider enhancing senior citizens’ access to religious and social gatherings. This could be accomplished by enhancing accessibility for religion and social gathering because mobility could become the biggest obstacle from the perspective of senior citizens. Second, they could allocate more resources towards increasing the jobs available for senior citizens, since economic activity exerted a greater impact on subjective health than participation in religious and social gatherings. Third, policy makers might consider providing senior citizens with financial support for food and leisure activity consumption, with more efficient allocations for leisure activity. It could be achieved by offering coupons for food and leisure activities for senior citizens. Finally, policy makers might contemplate budgeting for travel and art watching opportunities for senior citizens. Policy makers might be able to allot their resources for travel agency to make products for art watching and traveling focusing on senior citizens. Such budgeting might be able to improve the overall health condition of senior citizens, which in turn could lead to savings in social costs such as insurance and medical expenditure.

There are some limitations to this study. First, since we used secondary data, the measurement might have been less sophisticated. Future studies could develop advanced items for measurement to achieve a more precise estimation. Second, the sample in this study was limited to Korean senior citizens. Future studies could include a more diverse sample from several countries and compare the results with the current study. Such an effort could improve the generalizability of the determinants of subjective health for senior citizens. Plus, the data was a little bit outdated considering study period: 2018. Future research, thus, might be able to use more updated information which reflects the information of the corona virus disease 2019 outbreak.

## Figures and Tables

**Table 1 behavsci-12-00315-t001:** Variable description.

Name	Code	Description (Unit)
Subjective health	SH	(1 = Very bad, 5 = Very good)
Religion participation	RP	(0 = No, 1 = Yes)
Social gathering participation	SP	(0 = No, 1 = Yes)
Economic activity	EA	(0 = No, 1 = Yes)
Food expenditure	FE	Monthly food expenditure (10 thousand KRW)
Leisure expenditure	LE	Monthly leisure expenditure (10 thousand KRW)
Travel frequency	TF	Annual travel frequency (times)
Art watching frequency	AF	Annual art watching frequency (times)
Assets	AS	Personal assets (10 thousand KRW)
Age	AG	Age of survey participants
Gender	GE	(0 = Male, 1 = Female)

Note: KRW denotes Korean won.

**Table 2 behavsci-12-00315-t002:** Descriptive statistics (N = 3822).

Variable	Mean	SD	Minimum	Maximum
SH	2.98	0.88	1	5
RP	0.13	0.34	0	1
SP	0.63	0.48	0	1
EA	0.37	0.48	0	1
FE	42.59	25.02	0	220
LE	5.16	8.99	0	150
TF	1.18	2.15	0	40
AF	0.71	2.05	0	50
AS	20,425.93	32,703.69	0	788,000
AG	69.55	9.99	55	102
GE	0.42	0.49	0	1

Note: SD denotes standard deviation.

**Table 3 behavsci-12-00315-t003:** Correlation matrix (N = 3822).

Variable	1	2	3	4	5	6	7	8	9	10
1. SH	1									
2. RP	0.031 *	1								
3. SP	0.289 *	−0.146 *	1							
4. EA	0.308 *	−0.031 *	0.153 *	1						
5. FE	0.247 *	0.054 *	0.111 *	0.149 *	1					
6. LE	0.222 *	0.064 *	0.140 *	0.126 *	0.282 *	1				
7. TF	0.159 *	0.037 *	0.130 *	0.084 *	0.140 *	0.249 *	1			
8. AF	0.196 *	0.068 *	0.123 *	0.071 *	0.213 *	0.369 *	0.232 *	1		
9. AS	0.095 *	0.010 *	0.086 *	0.050 *	0.136 *	0.272 *	0.107 *	0.119 *	1	
10. AG	−0.440 *	−0.026 *	−0.255 *	−0.462 *	−0.374 *	0-.276 *	−0.188 *	−0.247 *	−0.066 *	1
11. GE	0.091 *	−0.096 *	0.036 *	0.241 *	0.032 *	0.043 *	0.023	−0.021	0.165 *	−0.034 *

Note: * *p* < 0.05.

**Table 4 behavsci-12-00315-t004:** Results of multiple regression analysis.

Variable	Coefficient	t-Value	*p*-Value
Intercept	4.267	34.46	0.000 *
RP	0.076	2.24	0.025 *
SP	0.271	10.26	0.000 *
EA	0.165	5.78	0.000 *
FE	0.002	3.94	0.000 *
LE	0.004	2.92	0.004 *
TF	0.013	2.53	0.012 *
AF	0.021	3.70	0.000 *
AS	0.001	3.25	0.001 *
AG	−0.024	−16.13	0.000 *
GE	0.105	3.80	0.000 *
F-value	121.08		0.000 *
R2	0.2411		

Note: Dependent variable: SH, SD denotes standard deviation, * *p* < 0.05.

**Table 5 behavsci-12-00315-t005:** Results of independent t-test.

Variable	Mean (No)	Mean (Yes)	t-Value	*p*-Value
RP	2.97	3.05	2.61	0.004 *
SP	2.64	3.17	25.18	0.000 *
EA	2.77	3.33	26.97	0.000 *

Note: Dependent variable: SH, SD denotes standard deviation, * *p* < 0.05.

## Data Availability

Not applicable.

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
