# Peer review of "Antecedents of Subjective Health among Korean Senior Citizens Using Archival Data"

_behavsci, 2022, doi:10.3390/bs12090315_

Round 1
Reviewer 1 Report
This paper is relevant because it deals with the determinants of subjective health among Korean senior citizens. Based on a review of the state of the art, the authors present a descriptive study using a very large sample and data collected by the Korea Employment Information Service. Other independent variables such as religion participation, social gathering participation, economic activity, food expenditure, leisure expenditure, travel frequency, and art watching frequency are also taken into account. The authors suggest that all these independent variables have a positive impact on the subjective health of Korean senior citizens. This is of particular interest for the development of policy proposals.
The main strengths of the manuscript are related to the chosen topic and its interest for the Korean society and elderly population. On the other hand, the design of the study is interesting as it includes a multiple linear regression analysis.
On the other hand, the theoretical framework of the study has the good sense to present an interesting review that gathers a good number of primary and secondary scientific sources. However, the authors are not correct in presenting only 12 references corresponding to the last five years out of more than 50 references cited. It is recommended that the percentage of references to publications from the last five years be increased to 40%. It is recommended that this percentage should increase considerably for papers published in the last 3 years. An out-of-date review has a negative impact on the discussion of the manuscript. It is recommended that the authors expand the discussion of the manuscript and that in the conclusions section they only refer to the achievement of the objectives of the study and the confirmation of the research hypotheses. It is also necessary that the authors indicate the limitations of their study and the future prospects of the research.
On the other hand, from a methodological point of view, it is necessary that the authors report on the size of the population and the procedure for recruiting participants: was it online, by telephone survey, by paper and pencil? In turn, the authors should review the participant section and indicate what incentives were used to recruit participants, if any. Also, whether ethical criteria for human subjects research were met.
.
Further details on the reliability and validity of the instrument used should also be provided in an 'instrument' section. Similarly, data on the type of method (design) followed in the study should appear in the "participants" section.
Overall, the paper is interesting but the review of the state of the art and the discussion and completeness of the data in the methodology section need to be substantially improved.
Author Response
Response paper
Manuscript ID: behavsci-1885136
Title: Antecedents of subjective health among Korean senior citizens using archival data
Reviewer 1
1.On the other hand, the theoretical framework of the study has the good sense to present an interesting review that gathers a good number of primary and secondary scientific sources. However, the authors are not correct in presenting only 12 references corresponding to the last five years out of more than 50 references cited. It is recommended that the percentage of references to publications from the last five years be increased to 40%. It is recommended that this percentage should increase considerably for papers published in the last 3 years. An out-of-date review has a negative impact on the discussion of the manuscript. It is recommended that the authors expand the discussion of the manuscript and that in the conclusions section they only refer to the achievement of the objectives of the study and the confirmation of the research hypotheses. It is also necessary that the authors indicate the limitations of their study and the future prospects of the research.
Response) Given the comments of reviewer, we updated the literature as more recent literature. Also, we attempted to strength the discussion part more.
- On the other hand, from a methodological point of view, it is necessary that the authors report on the size of the population and the procedure for recruiting participants: was it online, by telephone survey, by paper and pencil? In turn, the authors should review the participant section and indicate what incentives were used to recruit participants, if any. Also, whether ethical criteria for human subjects research were met.
Response) Give the comments of reviewer, we address the survey method at the manuscript. Also, since this research employed secondary data, the ethical issue is very little for human subject.
- Further details on the reliability and validity of the instrument used should also be provided in an 'instrument' section. Similarly, data on the type of method (design) followed in the study should appear in the "participants" section.
Response) Given the comments of reviewer, we address it at the measurement section. Moreover, since this research employed secondary data, all variables were measured by single item. It means that it is hard to apply the method for reliability and validity because variables are not multiple items.
- Overall, the paper is interesting but the review of the state of the art and the discussion and completeness of the data in the methodology section need to be substantially improved.
Response) Considering the comments of reviewer, we tried to strengthen the discussion and method part more. Additionally, we also made up for the introduction part more as well as discussion section.
Reviewer 2 Report
Review of Antecedents of subjective health among Korean senior citizens using archival data
1. Please order the section and sub sections accordingly to template, putting number on it.
2. Also citations in text need to be in order of appearance and with numbers in brackets.
3. References need to be numbered.
4. Check other behavioral science template topics and check if any other corrections are needed.
5. Better you put at abstract if it is north or south Korea…
6. Data from 2018 can be outdated nowadays. Despite human fundamental features remains the same for centuries, we should take care of small variations of behavior. To illustrate this context, take into account the COVID-19 influence on the human behavior. Regarding it, this reviewer ask authors, do these 2018 data have its validity to nowadays life linearly?
7. Concerning results and discussion, there are some points that needs clarification. We can check at table 2 that RP is quite low in terms of frequency and also table 3, its correlation to SH is very low despite we objectively know this variable has great influence on SH, therefore, if we follow the data, we should consider RP with very low influence on SH? Second point, we can check at table 2 EA, FE and LE have very low frequencies, which is certainly a negative trait to SH high score, despite people are saying like “3 (2.98)” points as mean. What would explain people saying “3” for an evident negative trait? Also in the conclusion section authors say RP, EA, FE and LE are important policies to be improved since they can influence higher SH scores, despite it is a common sense for us, but it can’t be extracted objectively from the data and results found.
8. I wonder how results of EA, FE and LE table 2 would behave themselves towards a period after COVID-19 like nowadays. If score of SH would be “3” really. I believe authors should state that this study was conducted in a period before COVID-19 alerting readers about this significant event that changed certainly the subjective health of people nowadays.
Author Response
Response paper
Manuscript ID: behavsci-1885136
Title: Antecedents of subjective health among Korean senior citizens using archival data
Reviewer 2
1.Please order the section and sub sections accordingly to template, putting number on it.
Response) Given the comment, we did the numbering for manuscript.
- Also citations in text need to be in order of appearance and with numbers in brackets.
Response) We will do it after the completion of revision because reviewers could give additional comments related to the reference. Please understand it.
- References need to be numbered.
Response) We will do it after the completion of revision because reviewers could give additional comments related to the reference. Please understand it.
- Check other behavioral science template topics and check if any other corrections are needed.
Response) We did check the format of behavioral science template, and tried to keep the template as much as possible.
- Better you put at abstract if it is north or south Korea…
Response) Considering the comment, we addressed South Korean at the abstract.
- Data from 2018 can be outdated nowadays. Despite human fundamental features remains the same for centuries, we should take care of small variations of behavior. To illustrate this context, take into account the COVID-19 influence on the human behavior. Regarding it, this reviewer ask authors, do these 2018 data have its validity to nowadays life linearly?
Response) We understand the point of reviewer. The information what we used in this work was the most recent piece. It might be useful comment for future research. Thus, we address it at the conclusion section for future research. The following is addressed at the conclusion section:
Plus, the data was little bit outdated considering study period: 2018. Future research thus might be able to use more updated information which reflects the information of corona virus disease 2019 outbreak.
- Concerning results and discussion, there are some points that needs clarification. We can check at table 2 that RP is quite low in terms of frequency and also table 3, its correlation to SH is very low despite we objectively know this variable has great influence on SH, therefore, if we follow the data, we should consider RP with very low influence on SH?
Response) Given the comment of reviewer, we addressed the point for RP at the discussion section.
8.Second point, we can check at table 2 EA, FE and LE have very low frequencies, which is certainly a negative trait to SH high score, despite people are saying like “3 (2.98)” points as mean. What would explain people saying “3” for an evident negative trait?
Response) Regarding the comment, we clarified the point at the discussion section as follows:
The results suggest that eating good food, traveling, and enjoying cultural activities are also very essential element for senior citizen to become healthier. In addition, the re-sults of independent t-test showed the following mean values (RP: 2.97 vs. 3.05; SP: 2.64 vs. 3.17; EA: 2.77 vs. 3.33). It showed that the mean value is between two and three. Considering the characteristics of survey participants (senior citizens), they ap-praised their subjective health relatively negative manners.
- Also in the conclusion section authors say RP, EA, FE and LE are important policies to be improved since they can influence higher SH scores, despite it is a common sense for us, but it can’t be extracted objectively from the data and results found.
Response) Given the comment of reviewer, we tried to strengthen policy implications.
- I wonder how results of EA, FE and LE table 2 would behave themselves towards a period after COVID-19 like nowadays. If score of SH would be “3” really. I believe authors should state that this study was conducted in a period before COVID-19 alerting readers about this significant event that changed certainly the subjective health of people nowadays.
Response) We understand the point of reviewer. The information what we used in this work was the most recent piece. It might be useful comment for future research. Thus, we address it at the conclusion section for future research. The following is addressed at the conclusion section:
Plus, the data was little bit outdated considering study period: 2018. Future research thus might be able to use more updated information which reflects the information of corona virus disease 2019 outbreak.
Round 2
Reviewer 1 Report
I believe that the authors have made the modifications I suggested fully and effectively. As a result, the quality of the article has been enhanced. In conclusion, I endorse the manuscript.
Reviewer 2 Report
Majority of comments were addressed properly. No more coments to add. Thank you to the authors.